

**ENSO-triggered floods in South America:**
**correlation between maximum monthly discharges during strong events**
Federico Ignacio Isla
Instituto de Geología de Costas y del Cuaternario (UNMDP-CIC)
Instituto de Investigaciones Marinas y Costeras (UNMDP-CONICET)
Funes 3350,  Mar del Plata 7600, Argentina, +54.223.4754060, fisla@mdp.edu.ar
**Abstract**
ENSO-triggered floods altered completely the annual discharge of many watersheds of South America. Anomalous
years as 1941, 1982-83, 1997-98 and 2015-16 signified enormous fluvial discharges draining towards the Pacific
Ocean, but also to the Atlantic. These floods affected large cities built on medium-latitudinal Andes (Lima, Quito,
Salta), but also those located at floodplains, as Porto Alegre, Blumenau, Curitiba, Asunción, Santa Fe and Buenos
Aires. Maximum discharge months are particular and easily distinguished along time series from watersheds located
at the South American Arid Diagonal. At watersheds conditioned by precipitations delivered from the Atlantic or
Pacific anti-cyclonic centers, the ENSO-triggered floods are more difficult to discern. The floods of 1941 affected
70,000 inhabitants in Porto Alegre. In 1983, Blumenau city was flooded during several days; and the Paraná River
multiplied 15 times the width of its middle floodplain. That year, the Colorado River in Northern Patagonia
connected for the last time to the Desagûadero – Chadileuvú - Curacó system and its delta received saline water for
the last time.  During strong ENSO years the water balances of certain piedmont lakes of Southern Patagonia are
modified as the increases in snow accumulations cause high water levels, with a lag of 13 months. The correlation
between the maximum monthly discharges of 1982-83 and 1997-98 at different regions and watersheds indicates
they can be forecasted for future floods triggered by same phenomena. South American rivers can be classified
therefore into ENSO-affected and ENSO-dominated for those within the Arid Diagonal that are exclusively subject
to high discharges during those years.
Keywords: floods, El Niño-Southern Oscillation, South America, maximum monthly discharges





## 1. Introduction


El Niño events were known before the Spanish colonized the Peru region because of their consequences on the
anchovy fishery. They were also known by seasonal heavy rainfalls and rapid floods in tropical South America.
However, Jules Verne exaggerated these flash floods as occurring on the Pampas plains in his book *Les enfantes du*
*Capitaine Grant* (1868, reproduced in 1962 in the Disney´s movie *In search of the Castaways)*. These floods do not
occur as rapid; they are the response of several weeks or months with rains over the average.
Their origin is well known: immense volumes of water transported across the Pacific Ocean during certain
anomalous years, the so-called "*El Niño''* or ENSO years (Vargas et al. 2000; Andreoli and Kayano 2005). Although
this interannual anomalous years are known by their climatic and oceanographic consequences, their hydrological
responses in South American rivers have not been carefully reported. One to the main reason is the lack of
information about rains records (Sun et al., 2015) and also of long and continuous hydrological records (Ward et al.
38 2016).

Rapid floods at the Andes watersheds occur during strong ENSO years, impacting in Peru, Ecuador and Northern
Chile. They are significantly recorded when they affect arid watersheds comprised within the South American Arid
Diagonal. However, these interannual floods also affect extended Atlantic watersheds of rivers as the Paraguay,
Bermejo, Pilcomayo and Salado. This manuscript reported the available records of these floods -in their monthly
periodicity-, compiled at national agencies of different countries (Ecuador, Peru, Chile, Paraguay, Brazil, Argentina
and Uruguay). Environmental and social impacts of these floods in South America were reported considering
specially that the floods triggered by El Niño-La Niña are significantly longer (Ward et al. 2016).

## 2. Climate


Central South America has a subtropical to temperate climate. Humidity is provided from the east by trade winds
from the anti-cyclonic center of the South Atlantic. Further south, humidity is also provided by westerly winds from
the South-Pacific anti-cyclonic center. Between both humid areas, the Arid South American Diagonal (ASAD)
extends from N to S, connecting the Atacama and Patagonian deserts (Fig. 1). Climate was considered as the main
variable governing the suspended sediment yields from catchments basins located to the E of the Andes between
Ecuador and Bolivia, either in its variability or indirectly conditioning the vegetation cover (Pepin et al. 2013).
Along the coast of Chile, rains increases from north to south (Valdés-Pineda et al. 2014; Araya Ojeda and Isla
2016). On the other hand, along the Eastern Patagonia coast, rains increase northwards (Coronato et al. 2008).

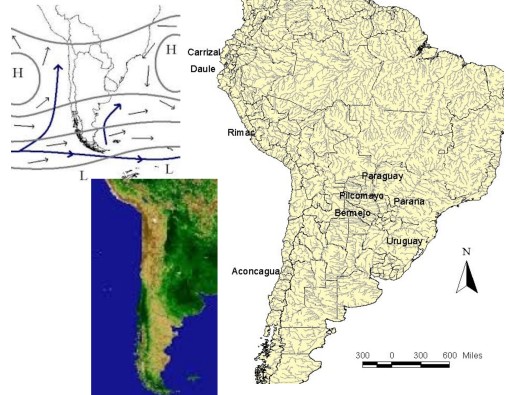






*Fig. 1. A. Anti-cyclonic centers ejecting winds from the east and west. B. South America is characterized by the*
*ASAD connecting Atacama and Patagonia deserts. C. Major rivers of South America.*
South America has significant temporal changes in its interannual precipitation. Precipitation has a linear response
to El Niño occurrences (Andreoli and Kayano 2005). In regard to long-term precipitation, south of 15°S, there were
positive jumps east of the Andes, with a negative trend toward the west (Minetti and Vargas 1997). Historical
positive jumps occurred between 1946 and 1960 while the negative trend diminished from north (Antofagasta
station) to south (Islote Evangelista meteorological station).
In Southern Chile, and according to records measured at Valdivia, there was a significant decrease in precipitations
between 1901 and 2005 (González-Reyes and Muñoz 2013). On the other coast, at the Argentine Pampas, there was
an increase in 50-200 m in the annual rains comparing two intervals: 1947-1976, and 1977-2006 (Forte Lay et al.
2008). The Pampa Region increases its Precipitation rates during the last decades of 20th century (Scarpatti and
Capriolo 2013). Several authors point to the early 70's as the epoch of significant increases in runoffs of the rivers
Paraguay and Paraná (Pasquini and Depetris 2007).Notwithstanding this natural climatic scheme, significant
variations in South America should be assigned to changes in the land use and land covers (Clark et al. 2012).

## 3.  Methods
Monthly hydrological records were compiled and analyzed from the databases of different South American
countries (Table 1).

| Peru | http://www.senamhi.gob.pe/ |
| Ecuador | http://www.serviciometeorologico.gob.ec/caudales-datos-historicos/ |
| Chile | http://snia.dga.cl/BNAConsultas/reportes |
| Brazil | http://hidroweb.ana.gov.br/HidroWeb.asp?TocItem=4100 |
| Argentina | http://www.mininterior.gov.ar/obras-publicas/rh-base.php |

*Table 1. Web pages of the hydrological records of different countries of South America.*
Historical maps and TM images of the Landsat satellites were compared in order to discriminate the extension of
flooding episodes from normal conditions.

## 4.  Results
### 4.1. Ecuador and Peru
According to the Ecuadorian INAMHI institution, the largest floods connected to strong El Niño phenomena
occurred in 1977-78, 1982-83 and 1997-98. The floods of 1983 were triggered by enormous amounts of rainfall at
Western Ecuador (Rossel et al. 1996).  The impacts caused by the strong ENSO of 1997-98 were estimated
according to different economic sectors: agriculture (43.6 MU$S), infrastructure of the sanitary sector (27.5 MU$S),
housing (3.2 MU$S) and industry (9.5 MU$S; Vaca 2010).





The Daule River (Fig. 1) was flooded in 1965, 1983, 1997-98 and 2012 (Fig. 2). Normally these maximum
discharges occurred during the first months of the year (January to May). The worst floods were during the first
months of 1998 with discharges over 1300 m³/s (Fig. 2).   At the boundary between Ecuador and Peru, the Zarumilla
River increased its discharge during the years 1965, 1973, 1983 and 1998.

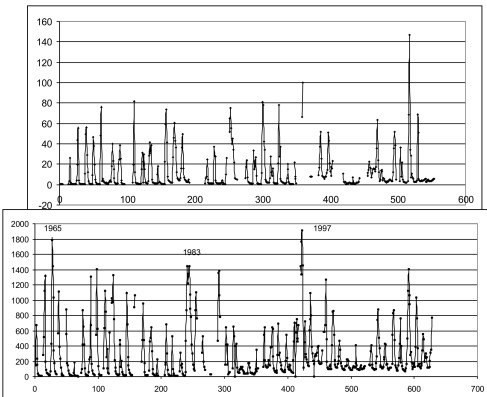


*Fig. 2. Monthly discharges of the Carrizal River (top) and Daule River in La Capilla showing peaks in 1965, 1983,*
*1997 and 2012.*
Perú hydrological statistics are published every year by the SENAMHI (2016). The Rimac River (Fig. 1) flooded in
1925 with a maximum daily discharge of 600 m³/s; an event considered the first "meganiño" of the 20[th] century
(Rocha Felices 2011). It was also flooded in 1941 (385 m3/s) and 1955 (380 m3/s). Historical data from Perú
indicates that there is a patchy distribution between different basins (Waylen and Caviedes 1986).

**4.2. Chile**
Although Chilean floods may occur by different origins, 71% are associated to rainfalls. However, rainfalls are
assumed to be diminishing in a long-term scenario (González-Reyes and Muñoz, 2013). Those floods associated
exclusively to strong ENSO events occur northwards of 36°S (Rojas et al. 2014). However, significant discharges
also occur at the south, but masked to other floods triggered by local rains (Araya Ojeda and Isla 2016).
There is not a definite effect of ENSO anomalies along the whole Chile. Those rivers of Northern Chile comprised
within the South American Arid Diagonal are specifically subject to anomalous precipitations. The two debris flow
recorded in Antofagasta  in 1940 (Vargas et al 2000) could have been also connected to the strong ENSO of
1941.The 1982-83 and 1997-98 ENSO rainfalls affected significantly Northern Chile (Meza 2013; Vargas et al.
107  2006).

In Central Chile, the higher discharges of the Aconcagua River (Fig. 1) were related to ENSO events but with a
certain delay (Waylen and Caviedes 1990). For the interval 1901-2005 there was a significant reduction of annual
precipitation for the Valdivia region, southern Chile (González-Reyes and Muñoz, 2013). It has been proposed that
the reduction in water yields in South-Central Chile is caused by land-use changes derived from the replacement of
native forest by exotics (Little et al. 2009); afforestation significantly affect runoff at the Biobio Region (Iroumé and
Palacios 2013).





### 4.3. Brazil

Anomalous years affected some cities of Brazil. The floods of 1941 affected 70,000 inhabitants at the riverine area
of Porto Alegre (Fig. 3). City authorities constructed a dike in order to prevent another flood of the Guaiba fluvial
complex (Loitzenbauer et al. 2012).

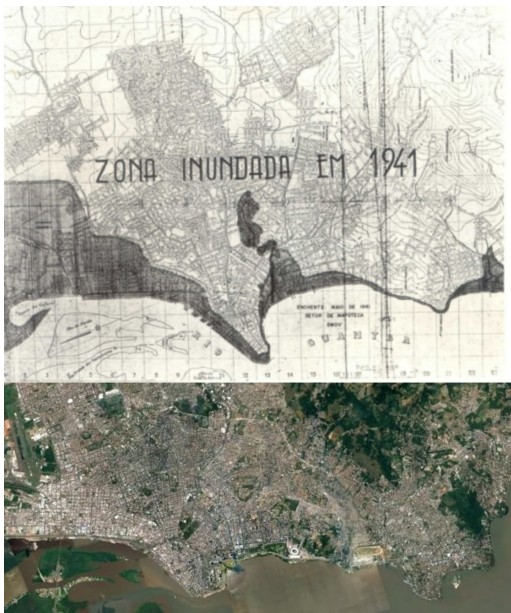


*Fig. 3. A) Fluvial area of Porto Alegre flooded in 1941. B) Present area.*

The floods of 1983 of the Itajaí-Açú River caused the destruction of 30,000 houses at Blumenau. The level of the
river raised 16 m over normal level and stayed high for several days. 80% of the Itajaí County was affected. The
Iguaçú River, an affluent to the Paraná River (Fig. 1), flooded Curitiba in 1982 and 1983 signifying losses of 10,000
and 78,000 MU\$S respectively at some neighborhoods (Tucci and Petry 2006).

### 4.4. Bolivia

The Pilcomayo River (Fig. 1) has a maximum discharge of 3500 m$^3$/s, about 45 times its minimum discharge (80
m$^3$/s; Rabicaluc 1986). This river has an alluvial fan of 210,000 km$^2$ with several abandoned channels (Iriondo et al.
2000). The floods of the Upper Pilcomayo River of 1983 and 1984 (Fig. 4) increased 2-3 times the amount of
sediment transported in suspension (Malbrunot 2006). During normal years the river transports less than 1 x 10$^6$ tons
of sediments; in 1984 it transported 2 and 3 millions of tons. The city of Villamontes (Tarija) is usually flooded by
the Pilcomayo River. Although a hydrologic gauge was installed in 1941, it operated randomly. A maximum level of
7.98 m was measured in March, 1984 with a maximum discharge of 7000 m$^3$/s (Ribstein and Peña 1993). The last
flood was recorded during the beginning of 2018.





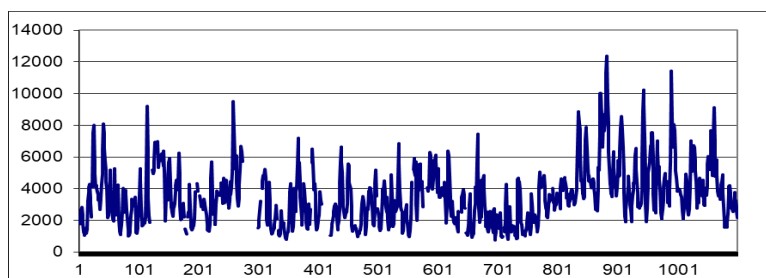


*Fig. 4. Pilcomayo River floods at Puerto Pilcomayo with the peaks in 1983, 1997 and 2015.*
The Bermejo River (Fig. 1) also flows from Bolivia to Argentina, to the Paraná River. Based on historical archives,
its hydrological cycles have been reconstructed (Prieto and Rojas 2015). Floods progressively increased since 1800.
The deforestation has increased the climatic effects. Floods frequency diminished during the first half of the 20th
century but increased significantly to the end of that century. These rivers that flow from the Andes to the Parana
River (Paraguay, Pilcomayo, Bermejo) carried significant amount of particulate and dissolved substances. The
plume of the Paraguay River persists isolated from the Upper Paraná water during approximately 225 km
(Campodónico et al. 2015).

**4.5. Paraguay**
The Republic of Paraguay is located between three rivers (Paraguay, Paraná and Pilcomayo), all belonging to the
Río de la Plata watershed (Baez et al. 2014). The Paraguay River (Fig. 1) is about 2800 km long draining an area of
about 1,095,000 km$^2$ (Collischonn et al. 2001). This large basin should be analyzed according to two regions: the
northern related to the Amazonas River system, and the southern, subject to ENSO-triggered floods (Drago et al.
2008). This watershed is in close relation to the Patiño Aquifer (Monte Domecq and Baez Benítez 2007) and the
Pantanal wetlands (Collischonn et al. 2001). During the winters of 1982 and 1983 the river had discharges of 9712
and 10663 m3/s, respectively (Monte Domecq et al. 2003; Barros et al. 2004). The two largest floods of the riverine
areas of Asunción occurred in 1983 (63 m over MSL) and 1992 (62.3 m). During the floods of 1997/98 24,975
inhabitants were evacuated from Asunción, and 54,000 inhabitants from other departments (Neembucú, Concepción,
Cordillera and Chaco). One of the major risks of the Paraguay River floods is that concerning to the Cateura waste
disposal of the Asunción city (Fig. 5). This dumping site is on the floodplain and very close to the international
boundary with Argentina.

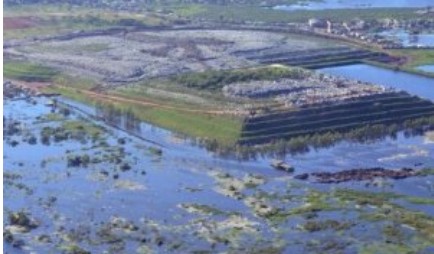


*Fig. 5. During the ENSO 2015-2016 the waste disposal site of Asunción was flooded very close to the international*
*boundary with Argentina.*




### 4.6. Argentina and Uruguay

The Rio de la Plata received significant amount of water from the Upper Paraná (83%), 20% from the Paraguay
River, and about 7% from the rivers flowing from the west (Bermejo, Pilcaomayo and Salado; Pasquini and
Depertris 2010). The Paraná River (Fig. 1) flooded systematically during the last strong ENSOs (1982-83, 1997-98
and 2015; fig. 6) It multiplied 15 times the widths of its floodplain during the floods of 1982-83 (Drago 1989). This
extraordinary event signified high monthly streamflows in Corrientes during a year and half (Camilloni and Barros
168 2003).

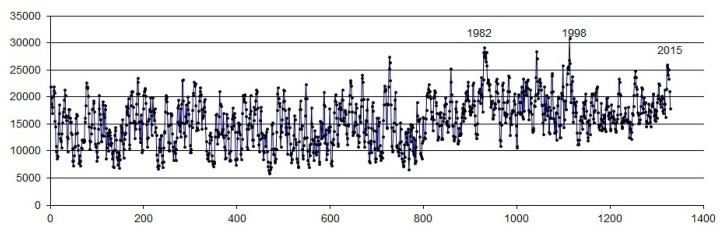


*Fig. 6. Hydrological record of the Paraná River at Timbúes Station (1901-2016).*
In Corrientes city, the discharge surpassed 10,000 m$^3$/s from July 1982 to December 1983 (Camilloni and Barros
2003), also affecting the localities of Resistencia, Barranqueras, Puerto Vilelas and Fontana (Fig. 7). The Paraná
River at Barranqueras reached the maximum 8.6 m level. Below the General Belgrano Bridge the discharge was
58,000 m$^3$/s.

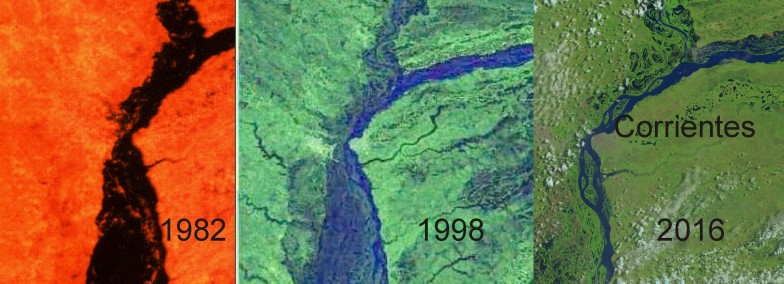


*Fig. 7. The floods of of 1982, 1998 and 2016 affected the cities of Corrientes and Resistencia (Argentina).*
These floods signified the transport of subtropical floating plants (*Eichhornia crassipes*, also known as "water
hyacinth") to temperate areas, and carrying dangerous fauna with them (snakes, spiders and lizards). Several fluvial
harbors as Rosario, Campana, Zárate and Buenos Aires were restricted in their operability during these events. At
the floodplain close to Rosario, the peak flows of 1982-83, 1992, and 1997-98 exceeded 30,000 m$^3$/s (Fig. 8).
During these extraordinary floods, the floodplain stores between 23 and 123 x 10$^6$ tons/year (García et al. 2015).



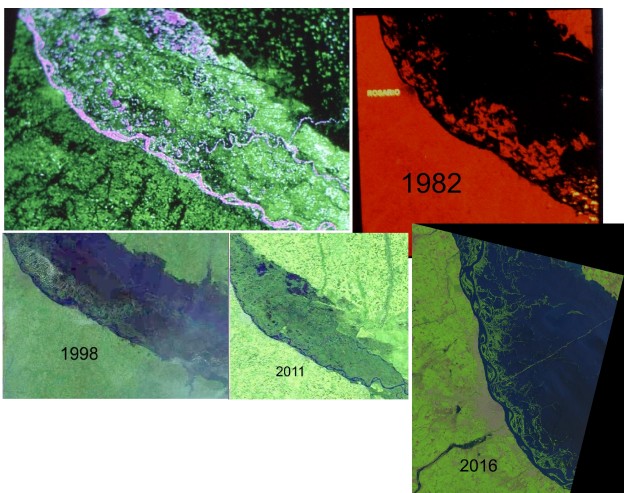


183 *Fig. 8. The Paraná River flooded several times restricting the operation of the harbor of Rosario.*

184 The Uruguay River (**Fig. 1**) flooded in 1941, 1983 and 1997-98 (Isla and Toldo 2013). It has a mean discharge of
185 4315 m3/s (Evarsa 2006). Harmonic analysis shows a dry period during the 1950-1960 decade, recorded also at the
186 Paraná watershed (Krepper et al. 2003).

187 The Colorado River (Northern Patagonia) is assumed to deactivate from the northern portion of the watershed
188 during the Holocene. During the floods of 1982-83, the whole watershed connected for the last time and saline water
189 arrived to the delta plain (Isla and Toldo 2013). Proglacial lakes of eastern Patagonia were also affected during
190 ENSO years: increments in the amount of snow during ENSO years produce high water levels of these lakes with a
191 lag of 13 months (Pasquini et al. 2008).

192 **5. Maximum floods**

193 Comparing the best recorded strong ENSOs (1982-83 and 1997-98) they produced similar maximum discharges in
194 Chile (Araya Ojeda and Isla 2016). The strong ENSO of 1997-98 was stronger in Ecuador (Daule and Carrizal
195 rivers). However, and comparing their maximum monthly discharges, these floods are correlated (Fig. 9). This
196 correlation is useful to forecast the maximum discharges expected for future strong ENSOs. During the last two
197 centuries, the three strong ENSOs occurred in less than 40 years (1982-82, 1997-98 and 2015-2016) and are
198 therefore indicating a higher frequency in regard to previous years.




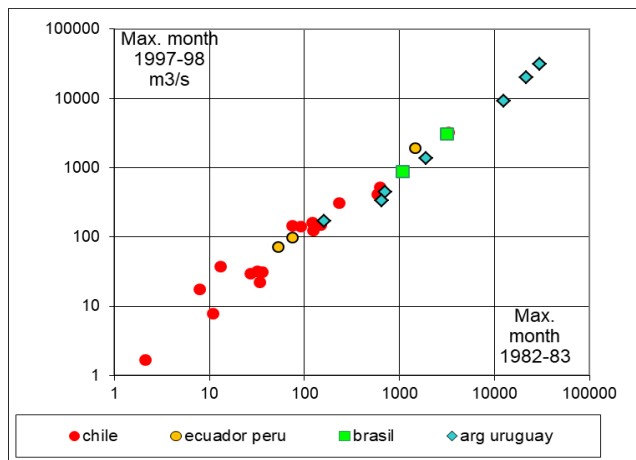

*Fig. 9. Comparison of maximum discharges (m3/s) of the floods of 1982-83 and 1997-98.*

**6.  Discussion**
The ENSO events recorded during the Holocene are highly controversial (Clement et al. 2000). Although this
review is based on hydrological measurements from countries with short series, there are some records that could be
applied as historical and tree-ring records. Paleoclimatic studies indicate that in Northern Chile (north of 30° S)
there was an absence of heavy rainfalls between 8400 and 5300 years BP in conjunction to a decrease in ENSO
activities at the Eastern Pacific Ocean (Ortega et al. 2012). Based on tree rings from the Bermejo River region, it
was stated that for the last three centuries there was significant increments in the frequency, intensity and duration of
floods and droughts since the second half of the 20[th] century (Ferrero et al. 2015). The last five extreme wet events
occurred since 1814, the last three in the last 40 years. However, there were significant droughts in Western Pampas:
the "Pampas Dust Bowl" occurred between 1930 and 1940 (Viglizzo and Clark 2006). Summarizing, for
Northeastern Argentina 1901-1960 was a dry period while 1970-2003 was characterized by wet conditions (Lovino
et al. 2014). ENSO floods occur with a different delay between the high-relief Andes watersheds draining towards
the Pacific Ocean and those meandering towards the Atlantic Ocean. In Patagonia, the delay between the snow
recharge and the raise in the piedmont lakes levels is about 13 months (Pasquini et al. 2008).
ENSO cycles do not only affect the hydrological records of South America. They also affect rivers of China causing
variations in their sediment discharge (Liu et al. 2017), and can therefore considered a good predictor for flood-
affected and flood-destroyed crop areas (Zhang et al. 2016).
ENSO cycles, either Niños or Niñas, have significant effects on the global price of wheat.  Niños cause reductions of
1.4% in its production while Niñas cause reductions of 4% (Ubilava 2017). Niños have positive effects regarding
crop yields at the Argentine Pampa; maize and wheat yields increase during ENSOs, while the increase in soybean
only occurred along some areas (Magrin et al. 1998). On the other hand, sunflower yields diminish during ENSO
years. These increments are more significant in the north of the Pampas region (Fernández Long et al. 2011).

**7.  Conclusions**



1. Strong ENSO floods affected South America in 1941, 1982-83, 1997-98 and 2015-16.

2. Rivers from the South American Arid Diagonal are only affected by ENSO floods. Those outside the diagonal can be also affected by anomalous precipitations derived from the Atlantic or Pacific oceans.

3. Comparing the monthly discharges of several rivers, the 1982-82 and 1997-98 floods were of similar magnitudes and should be considered to forecast future strong events and to organize mitigating plans.

**Acknowledgements**

The national services of Ecuador, Peru, Chile, Brazil and Argentina facilitated the monthly-collected data. An abstract of this paper was published during the EGU 2016, Vienna. This is a contribution to the floods project of Pages.

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
