# Peer review of "Discussion started: 3 April 2018"

_Hydrology and Earth System Sciences, 2018_

## Referee Comment (RC1) · Anonymous Referee #1 · 3 Apr 2018

An analysis of ENSO-triggered floods in South America is very welcome. Not only are these floods an extremely interesting geophysical phenomenon, they are also of utmost societal importance. The paper has access to an excellent data base that would be of interest to a wide readership, including scientists outside South America. Unfortunately, the paper reads more like a data report. For it to become publishable, I would encourage the authors to further explore the mechanisms that lead to these extraordinary floods, both in the atmosphere and on the land surface. While it is of interest to learn which floods have been affected by ENSO and which have not, this does not suffice for a scientific publication. An understanding of the mechanism of these floods, in particular the controls on their magnitude, would be necessary. Additionally, all line

graph figures are of poor quality and need to be redrawn with proper labels of the axes, and the grammar needs improving. I encourage the authors to resubmit a suitablly revised manuscript.

———————————————————

---

## Referee Comment (RC2) · R. Uijlenhoet (Referee) · 7 May 2018

R. Uijlenhoet (Referee)

remko.uijlenhoet@wur.nl

The author's main aim is to link the occurrence of major river floods in South America to the El Nino - Southern Oscillation (ENSO) phenomenon. This is done by analyzing time series of monthly discharges from major rivers in several South American countries. The most interesting result is Fig. 9, where the author shows a strong correlation between monthly peak discharges during two subsequent ENSO periods for the analyzed time series. This suggests that such peak discharges exhibit a high degree of predictability, conditional on the predictability of the ENSO phenomenon, which is an interesting observation with practical applicability.

However, the presentation of the data and methods, the results as well as the discussion in the current version of the manuscript lack the depth and rigor that are needed to warrant publication of the paper in HESS in its present form. Therefore, I am inclined to recommend rejection of the paper in it's current form, with an encouragement to resubmit a new version of this paper. In the attached annotated version of the manuscript, I provide a large number of detailed comments and suggestions, both concerning the contents of the paper and concerning the text.

My overall impression is that the paper in its current form is quite descriptive in its approach. This would make it difficult if not impossible for others to understand exactly how the reported results were obtained. Please try to be as exact as possible as to where the employed data may be found (also by others), what the quality of the data is (both in terms of data gaps and in terms of the discharge estimates themselves), and finally what methods were used to quality control and process the data.

Also, the idea of investigating the predictability of peak discharges by linking them to the ENSO phenomenon, interesting as it may be from a hydrological perspective, is currently not really investigated quantitatively. Although the employed data records are relatively short, I have the impression that a more quantitative appreciation of the mentioned predictability is actually possible. Perhaps most importantly, a discussion of the physical cause of the reported correlation is largely lacking.

Finally, the paper comes with a significant number of rather anecdotal statements concerning past floods (as well as some droughts) and their presumed relation to the ENSO phenomenon. Although they provide support for the claim that the link between ENSO and peak discharges is indeed existent in South America, many of the statements distract from the main storyline of the paper. Perhaps these can be summarized in a table, which would leave more room for more detailed presentations of the data, the methods, the results, the discussion and the conclusions.

Please see the comments and suggestions in the annotated paper for further details.

Please also note the supplement to this comment:
https://www.hydrol-earth-syst-sci-discuss.net/hess-2018-107/hess-2018-107-RC2-supplement.pdf

**Supplement:**

[revised manuscript text omitted]

---

## Author Comment (AC1) · 9 May 2018

The referee Uijlenhoet understood perfectly the virtues and deffects of the manuscript.

The correlation of the monthly peak discharges is the main contribution (the figure 9).

Methods descriptions can be improved. The results are difficult to extend as the hydrological records from some countries of South America have been discontinued. Many stations were abandonned after being destroyed several times during floods. In some countries the length of the records is certainly very short. although the importance of them is crucial for social purposes.

[Figure]

I will improve the manuscript focusing on the quality of the data and the physical cause of the correlation found (both suggested)

Please, let me know when I should upload a new version.

Best wishes

Federico Isla

Please also note the supplement to this comment:
https://www.hydrol-earth-syst-sci-discuss.net/hess-2018-107/hess-2018-107-AC1-supplement.pdf

---

## Author Comment (AC2) · 4 Jun 2018

1. An introduction to the ENSOs origins and their recurrence was added, as requested.
2. Dr. Uijlenhoet requests: a. An explanation about the quality of the data was added.
b. At the same time, the probable physical causes of the correlation of the maximum monthly discharges were discussed.

Figures were corrected and the suggestions of Dr. Uijlenhoet were all accepted.

A new version (ENSO in SA 9.pdf) was attached with many corrections (for example, the Bolivian web page for hydrological data was introduced) and the discussion was

improved

Please also note the supplement to this comment:
https://www.hydrol-earth-syst-sci-discuss.net/hess-2018-107/hess-2018-107-AC2-
supplement.pdf

—————————————————
107, 2018.

**Supplement:**

| 1 | ENSO-triggered floods in South America:                                      |
|---|------------------------------------------------------------------------------|
| 2 | correlation between maximum monthly discharges during strong events          |
| 3 | Federico Ignacio Isla                                                        |
| 4 | Instituto de Geología de Costas y del Cuaternario (UNMDP-CIC)                |
| 5 | Instituto de Investigaciones Marinas y Costeras (UNMDP-CONICET)              |
| 6 | Funes 3350, Mar del Plata 7600, Argentina, +54.223.4754060, fisla@mdp.edu.ar |
| 7 |                                                                              |
| 8 | bstract                                                                      |

ENSO-triggered floods altered completely the annual discharge of many watersheds of South America. Anomalous 10 years as 1941, 1982-83, 1997-98 and 2015-16 signified enormous fluvial discharges draining towards the Pacific 11 Ocean, but also to the Atlantic. These floods affected large cities built on medium-latitudinal Andes (Lima, Quito, 12 Salta), but also those located at floodplains, as Porto Alegre, Blumenau, Curitiba, Asunción, Santa Fe and Buenos 13 Aires. Maximum discharge months are particular and easily distinguished along time series from watersheds located 14 at the South American Arid Diagonal. At watersheds conditioned by precipitations delivered from the Atlantic or 15 Pacific anti-cyclonic centers, the ENSO-triggered floods are more difficult to discern. The floods of 1941 affected 16 70,000 inhabitants in Porto Alegre. In 1983, Blumenau city was flooded during several days; and the Paraná River 17 multiplied 15 times the width of its middle floodplain. That year, the Colorado River in Northern Patagonia 18 connected for the last time to the Desagûadero - Chadileuvú - Curacó system and its delta received saline water for 19 the last time. During strong ENSO years the water balances of certain piedmont lakes of Southern Patagonia are 20 modified as the increases in snow accumulations cause high water levels, with a lag of 13 months. The correlation 21 between the maximum monthly discharges of 1982-83 and 1997-98 at different regions and watersheds indicates 22 they can be forecasted for future floods triggered by same phenomena. South American rivers can be classified 23 therefore into ENSO-affected and ENSO-dominated for those within the Arid Diagonal that are exclusively subject 24 to high discharges during those years.

Keywords: floods, El Niño-Southern Oscillation, South America, maximum monthly discharges

**26 1. Introduction**

El Niño events were known before the Spanish colonized the Peru region because of their consequences on the anchovy fishery. They were also known by seasonal heavy rainfalls and rapid floods in tropical South America.

Their origin is well known: immense volumes of water transported across the Pacific Ocean during certain anomalous years, the so-called "*El Niño*" or ENSO years (Vargas et al. 2000; Andreoli and Kayano 2005).

Bjerknes (1969) postulated that ENSO originates when Sea Surface Temperatures (SST) anomalies in the Pacific

Ocean cause trade winds to strengthen or slacken driving ocean circulation changes that induce changes in the

**33** SST. Although much information has been collected from different projects (TOGA, TAO-TRITON, ECMWF)

there is still some doubts about the interactions that triggered ENSO events (Kleeman and Moore 1997; Neelin et al.

1998; Sheinbaum 2003, Dijkstra 2006). Although this interannual anomalous years are known by their climatic and oceanographic consequences, their hydrological responses in South American rivers have not been carefully

- 37 reported. One to the main reason is the lack of information about rains records (Sun et al., 2015) and also of long
- **38** and continuous hydrological records (Ward et al. 2016).

**39** Rapid floods at the Andes watersheds occur during strong ENSO years, impacting in Peru, Ecuador and Northern

Chile. They are significantly recorded when they affect arid watersheds comprised within the South American Arid

Diagonal. However, these interannual floods also affect extended Atlantic watersheds of rivers as the Paraguay,

Bermejo, Pilcomayo and Salado. This manuscript reports the available records of these floods -in their monthly periodicity-, compiled at national agencies of different countries (Ecuador, Peru, Chile, Paraguay, Brazil, Argentina and Uruguay). Environmental and social impacts of these floods in South America were reported considering especially that the floods triggered by El Niño-La Niña are significantly important (Ward et al. 2016).

**47 **2.** Climate**

Central South America has a subtropical to temperate climate. Humidity is provided from the east by trade winds from the anti-cyclonic center of the South Atlantic. Further south, humidity is also provided by westerly winds from the South-Pacific anti-cyclonic center. Between both humid areas, the Arid South American Diagonal (ASAD)

extends from N to S, connecting the Atacama and Patagonian deserts (Fig. 1). Climate is considered as the main variable governing the suspended sediment yields from catchments located to the E of the Andes between Ecuador and Bolivia, either in its variability or indirectly conditioning the vegetation cover (Pepin et al. 2013). Along the coast of Chile, rainfall increases from north to south (Valdés-Pineda et al. 2014; Araya Ojeda and Isla 2016). On the other hand, along the Eastern Patagonia coast, rainfall increases northwards (Coronato et al. 2008).

**Fig. 1. A. Anti-cyclonic centers ejecting winds from the east and west. B. South America is characterized by the ASAD connecting Atacama and Patagonia deserts. C. Major rivers of South America.**

South America has significant temporal changes in its interannual precipitation. Precipitation has a linear response to El Niño occurrences (Andreoli and Kayano 2005). With regard to long-term precipitation, south of 15°S, there were positive jumps east of the Andes, with a negative trend toward the west (Minetti and Vargas 1997). Historical

- 63 positive jumps occurred between 1946 and 1960, while the negative trend diminished from north (Antofagasta
- 64 station) to south (Islote Evangelista meteorological station; Fig. 1B).
- 65 In Southern Chile, and according to records measured at Valdivia (Fig. 1B), there was a significant decrease in precipitations between 1901 and 2005 (González-Reyes and Muñoz 2013). On the other coast, at the Argentine

Pampas, there was an increase of 50-200 mm in the annual rains comparing two intervals: 1947-1976, and 1977-

2006 (Forte Lay et al. 2008). The Pampa Region reports increments in precipitation rates during the last decades of the 20th century (Scarpatti and Capriolo 2013). Several authors point to the early 70's as the epoch of significant increases in runoffs of the rivers Paraguay and Paraná (Pasquini and Depetris 2007). Notwithstanding this natural

- 71 climatic influence, significant variations in South America should be assigned to changes in land use and land
- 72 covers (Clark et al. 2012).

**74 3. Methods**

Monthly hydrological records were compiled and analyzed from the databases of different South American countries (Table 1). As some of these hydrological stations are located in areas subject to floods, there are some gaps in coincidence with strong ENSO events. In this sense, the quality of the data is difficult to discern for each country and region. Some countries were successfully to achieve data bases with monthly records that can be downloaded in

- 79 digital formats; other countries have their records as printed tables. From some areas, there are automatic gauges
- 80 while some systems take several months to disseminate the information. Some countries require specific
- 81 permissions from the administration authorities.

| Peru      | http://www.senamhi.gob.pe/                                         |
|-----------|--------------------------------------------------------------------|
| Ecuador   | http://www.serviciometeorologico.gob.ec/caudales-datos-historicos/ |
| Chile     | http://snia.dga.cl/BNAConsultas/reportes                           |
| Brazil    | http://hidroweb.ana.gov.br/HidroWeb.asp?TocItem=4100               |
| Bolivia   | http://www.senamhi.gob.bo/web/public/sige                          |
| Argentina | http://www.mininterior.gov.ar/obras-publicas/rh-base.php           |

*Table 1. Web pages for hydrological records of different countries of South America.*

discriminate the extension of flooding episodes from normal conditions.

Historical maps and TM images (30 m pixel resolution) from Landsat satellites were compared in order to

**86 4. Results**

**87 4.1. Ecuador and Peru**

According to the Instituto Nacional de Meteorología e Hidrología from Ecuador, the largest floods connected to strong El Niño phenomena occurred in 1977-78, 1982-83 and 1997-98. The floods of 1983 were triggered by enormous amounts of rainfall at Western Ecuador (Rossel et al. 1996). The impacts caused by the strong ENSO of
 1997-98 were estimated for different economic sectors: agriculture (43.6 MU\$S), infrastructure of the sanitary sector (27.5 MU\$S), housing (3.2 MU\$S) and industry (9.5 MU\$S; Vaca 2010).

- 93 The Daule River (Fig. 1) produced floods in 1965, 1983, 1997-98 and 2012 (Fig. 2). Normally these maximum
- 94 discharges occur during the first months of the year (January to May). The worst floods occurred during the first
- 95 months of 1998. with discharges over 1300 m3 s-1 (Fig. 2). At the boundary between Ecuador and Peru, the
- **96** Zarumilla River produced discharge peaks during the years 1965, 1973, 1983 and 1998.

- Fig. 2. Monthly discharges of the Carrizal River (top) and Daule River in La Capilla showing peaks in 1965, 1983,
and 2012.
- 100 Perú hydrological statistics are published every year by the SENAMHI (2016). The Rimac River (Fig. 1) flooded in
- 101 1925 with a maximum daily discharge of  $600 \text{ m}^3 \text{ s}^{-1}$ , an event considered the first "meganiño" of the  $20^{\text{th}}$  century
- 102 (Rocha Felices 2011). It also produced a flood in 1941 ( $385 \text{ m}^3 \text{ s}^{-1}$ ) and 1955 ( $380 \text{ m}^3 \text{ s}^{-1}$ ). Historical data from Perú
- 103 indicates that there is a patchy distribution between different basins (Waylen and Caviedes 1986).
- 104

**105 4.2. Chile**

Although Chilean floods may have different origins, 71% are associated with rainfalls. However, rainfall is assumed to be diminishing in a long-term scenario (González-Reyes and Muñoz, 2013). Floods associated exclusively with strong ENSO events occur northwards of 36°S (Rojas et al. 2014). However, significant discharges also occur at the south, but masked to other floods triggered by local rains (Araya Ojeda and Isla 2016).

There is not a definite effect of ENSO anomalies along the whole Chile. Those rivers of Northern Chile comprised within the South American Arid Diagonal are specifically subject to anomalous precipitations. The two debris flow recorded in Antofagasta in 1940 (Vargas et al 2000) could have been connected to the strong ENSO of 1941. The

1982-83 and 1997-98 ENSO-related rainfalls significantly affected Northern Chile (Meza 2013; Vargas et al. 2006).

- 114 In Central Chile, the higher discharges of the Aconcagua River (Fig. 1) were related to ENSO events, but with a
- certain delay (Waylen and Caviedes 1990). For the interval 1901-2005 there was a significant reduction of annual
- precipitation for the Valdivia region, southern Chile (González-Reyes and Muñoz, 2013). It has been proposed that
- the reduction in water yields in South-Central Chile was caused by land-use changes due to the replacement of
- 118 native forest by exotics (Little et al. 2009); afforestation significantly affected runoff at the Biobio Region (Iroumé
- **119** and Palacios 2013).
- 120

**121 4.3. Brazil**

- 122 Anomalous years affected some cities of Brazil. The floods of 1941 affected 70,000 inhabitants at the riverine area
- of Porto Alegre (Fig. 3). City authorities constructed a dike in order to prevent another flood of the Guaiba fluvialcomplex (Loitzenbauer et al. 2012).